# Artificial Neural Network Model for Membrane Desalination: A Predictive and Optimization Study

MieowKee Chan [1,*], Amin Shams [2], ChanChin Wang [3], PeiYi Lee [1], Yousef Jahani [4] and Seyyed Ahmad Mirbagheri [5]

[1] Centre for Water Research, Faculty of Engineering, Built Environment and Information Technology, SEGi University, Jalan Teknologi, Kota Damansara, Petaling Jaya 47810, Selangor, Malaysia

[2] Department of Civil and Environmental Engineering, Faculty of Civil Engineering, Semnan University, Semnan 35131-19111, Iran

[3] Centre for Modelling and Simulation, Faculty of Engineering, Built Environment and Information Technology, SEGi University, Jalan Teknologi, Kota Damansara, Petaling Jaya 47810, Selangor, Malaysia

[4] Department of Plastic, Faculty of Processing, Iran Polymer and Petrochemical Institute, Pajoohesh Blvd, District 22, Tehran 14977-13115, Iran

[5] Department of Civil and Environmental Engineering, K. N. Toosi University of Technology, No. 1346, Vali Asr Street, Mirdamad Intersection, Tehran 19697-64499, Iran

* Correspondence: mkchan@segi.edu.my

**Abstract:** Desalination is a sustainable method to solve global water scarcity. A Response Surface Methodology (RSM) approach is widely applied to optimize the desalination performance, but further investigations with additional inputs are restricted. An Artificial neuron network (ANN) method is proposed to reconstruct the parameters and demonstrate multivariate analysis. Graphene oxide (GO) content, Polyhedral Oligomeric Silsesquioxane (POSS) content, operating pressure, and salinity were combined as input parameters for a four-dimensional regression analysis to predict the three responses: contact angle, salt rejection, and permeation flux. Average coefficient of determination ($R^2$) values ranged between 0.918 and 0.959. A mathematical equation was derived to find global max and min values. Three objective functions and three-dimensional diagrams were applied to optimize effective cost conditions. It served as the database for the membranologists to decide the amount of GO to be used to fabricate membranes by considering the effects of operating conditions such as salinity and pressure to achieve the desired salt rejection, permeation flux, contact angle, and cost. The finding suggested that a membrane with 0.0063 wt% of GO, operated at 14.2 atm for a 5501 ppm salt solution, is the preferred optimal condition to achieve high salt rejection and permeation flux simultaneously.

**Keywords:** ANN; contact angle; desalination; salt rejection; permeation flux

## 1. Introduction

In the 21st century, the world will face increased pressure from energy and water scarcity due to climate change and continuous population growth [1]. To address the concern, two strategies have been implemented. The first is an eco-friendly restoration of contaminated sites [2]. The other is to desalinate seawater to obtain fresh water from the vast and inexhaustible marine resources [3]. Given the enormous energy needs of direct distillation, reverse osmosis (RO) membrane technology, with 44% of the world's freshwater desalination, is currently the most essential approach to obtaining fresh water from the sea/sullied water [4]. Membrane properties, chemical composition, and surface properties affect the ability to remove contaminants, permeability, and antifouling properties [5]. Therefore, significant efforts have been made to improve their properties by adding additives to the membrane [6].

Among the several approaches used in the study to change the material and composition of the membrane, the incorporation of nanomaterials for the production of nanocomposite membranes appears to be a promising method for improving the filtration performance of membranes [7]. Therefore, significant efforts have been made to improve their properties by adding additives to the membranes [8]. Thus far, various types of hydrophilic nanoparticles, such as aluminum oxide ($Al_2O_3$) [9], carbon nanotubes (CNTs) [10], silica ($SiO_2$) [11], zeolite [12], zinc oxide (ZnO) [13], metal oxide nanoparticles [14–16], and $TiO_2$ [17], have been used to combine with membranes to improve their performance. Graphene oxide (GO) is one of the most favorable and widely used nanoparticles with its intrinsic properties such as having an extraordinary surface area, bacterial resistance, and high chemical and mechanical strength [18]. The excellent properties of GO are also reflected in GO/polymer nanocomposites [19] because of the hydrophilic functional groups (e.g., hydroxyl, carboxyl, and epoxy groups) in GO nanoparticles. They also exhibit high compatibility with polymer matrices via covalent or non-covalent bonds [20] and are generally easier to disperse in polar solvents or water than other nanomaterials [6].

Incorporating GO into the membrane matrix improves the hydrophilicity of the membrane surface, improving water permeability, and membrane performance [21]. The research findings from Shams et al. [19,21] showed that factors such as GO content, Polyhedral Oligomeric Silsesquioxane (POSS) content, operating pressure, and salinity were significant factors affecting membrane performance in terms of salt rejection and permeation flux. The quadratic models developed using Response Surface Methodology (RSM) showed an $R^2$ value within the range of 0.96–0.98. However, the quadratic models were developed based on only three important factors, which were GO content, salinity, and operating pressure in the first paper [19] and GO content, POSS content, and operating pressure in the second paper [21]. Due to the limitation of RSM, where the post-experiment analysis could not be done by merging the findings of the previous works, Artificial Neuron Network (ANN) was introduced in this study to develop a comprehensive predictive model for permeation flux, salt rejection, and contact angle by including all the factors, namely GO content, salinity, operating pressure, and POSS content.

Incorporating machine learning into membrane and separation research helps optimize the process and predict membrane separation performance. Waqas et al. [22] developed the models to optimize the membrane permeability in a membrane rotating biological contactor. The operating parameters, such as the disk rotational speed, hydraulic retention time, and sludge retention time, were optimized in the study. The result showed that the ANN model exhibited a $R^2$ value of 0.999 compared to the Support Vector Machine (SVM) grid search and SVM random search, which were 0.983 and 0.989, respectively. In recent work done by Behnam et al. [23], in predicting the performance of a direct contact membrane distillation module, they found that machine learning tools such as ANN and Support Vector Regression (SVR) predicted the permeate flux better than the mechanistic model with a lower computational cost. The statistical data suggested that the ANN model was better than SVR with a lower mean absolute percentage error, which was 3.46% compared to 4.78% for SVR. Kovacs et al. [24] compared the performance of different machine learning techniques, for instance, random forest (RF), ANN, and long-short term memory network (LSTM), in predicting the transmembrane pressure (TMP), which is an important parameter in membrane fouling studies. The result showed that the R2 of the ANN model was within 0.974–0.910, which was higher than the LSTM, which was 0.935–0.883. Meanwhile, the RF model was found to be good at predicting the extreme TMP values.

ANN showed good predictive performance as indicated in the previous studies, as stated above, as it is capable of establishing non-linear multivariate analysis given the empirical datasets of the specific regions for the training. ANN is widely used in chemical engineering-related research, such as predicting and (or) optimizing the performance of glucose production, biogas production [25], and distillation processes [26]. In the area of desalination, Mahadeva et al. [27] developed an ANN model to predict the performance of an RO-based desalination process. The permeate flux was predicted using the condenser

and evaporator inlet temperatures, flow rate, and salinity as inputs. The results showed that a model with two hidden layers of 20 nodes each, the softmax-purelin function as the output layer, and the Levenberg–Marquardt training function demonstrated the best prediction performance, where $R^2 = 0.9947$ and a mean square error (MSE) of 0.003 were achieved. Recent work done by Adda and team [28] in modeling the desalination process using ANN showed that the predictions of permeate conductivity, permeate flowrate, and recovery were in good agreement with the experimental data. The R correlations of these outputs were within the range of 0.94–0.96, which was higher compared to the model developed using a multi-linear regression model, where the R correlations were between 0.57 and 0.68. This shows that ANN is a reliable tool for accurately modeling the membrane separation process.

According to the latest review published by Jawad et al. [25] and Behnam et al. [29] on modeling the desalination process, the models were mainly developed by using operating conditions such as feed concentration, transmembrane pressure, and cross flow velocity as the inputs to predict flux and rejection rate. Although the membrane properties are vital in the separation process, the membrane materials are rarely adopted as one of the inputs in developing the model. Lately, the studies done by Mohd Amiruddin [30], Chan [31], and Chan and Ng [32] reported that the contact angle could be predicted from the membrane properties, which were closely related to the membrane formulation. However, the membrane separation membrane performance was not included in the study.

In order to provide a comprehensive study by considering both the membrane material and operating condition in modeling, the objective of this study is to develop an ANN model to predict the membrane desalination performance by considering the factors that govern the membrane separation performance, namely GO content, POSS content, operating pressure, and salinity. The contact angle, which reveals the hydrophilicity of the membrane, the salt rejection, and the permeation flux are the outputs. The membrane performance is optimized, and all the possible solutions under the optimized condition are calculated and visualized using the novel 3D wireframe block. Lastly, a cost analysis for varied scenarios is introduced for the first time in this study to recommend the most cost-effective membrane formulation for desalination purposes. This cost analysis allows the membranologists to tailor the membrane to meet the specific requirements of the users, such as high salt rejection with low permeation flux, high permeation flux with moderate salt rejection, or moderate salt rejection and permeation flux.

## 2. Materials and Methods

### 2.1. Data Collection

The ANN models were trained using the experimental data obtained from past research works [19,21], as shown in Table 1. The training dataset features the merger of two different sets of experimental data in Box–Behnken (BBD) [21] and Central Composite Design (CCD) [19] structures, where the purpose is to consolidate the early investigations by RSM on the desalination process using the advantages of the ANN technique. The dataset forms a union of the 13-point BBD and 15-point CCD to have 28 points in total, thus allowing the analytical model to expand from three to four dimensions of input variables. The proposed ANN model incorporated GO content (wt%), POSS content (wt%), salinity (ppm), and operating pressure (atm) as input variables to predict the outputs for contact angle (°), salt rejection (%), and permeation flux ($L/m^2h$). GO and POSS are the additives used in membrane fabrication to enhance the membrane properties such as flux and hydrophilicity [33,34].

**Table 1.** ANN datasets.

| ANN Dataset | Number of Data | Source | Experimental Data |
|---|---|---|---|
| Training and Validation | 28 | [19,21] | DoE (BBD + CCD) |

Since the experimental data were attained through the Design of Experiments (DoE) procedure, where each measuring point was pre-defined uniquely to form BBD and CCD structures, these points are essential for regression analysis to bound the entire domain in training the ANN models. Hence the K-fold cross-validation scheme with a 70/30% split for training and validation datasets in 3 folds was adopted to follow the fitting evaluation procedure.

## 2.2. Artificial Neural Network Modeling

In this study, we used 'MIKA.NN' for the ANN modeling and training. MIKA.NN is an analytical tool developed by the authors for engineering research applications with built-in pre- and post-processors capable of handling multivariate analysis [35], where the four types of data scaling methods—linear 1 (from 0 to 1), linear 2 (from −1 to 1), sigmoid, and z-score—can be made. The Stochastic Gradient Descent (SGD) algorithm is adopted in the backpropagation for training the neural network. Data normalization and standardization scaling methods were applied to cast the raw data of different units into isoparametric ranges [from −1 to +1] and [from 0 to 1] to eliminate influencing factors and stabilize errors in backpropagation. In addition, the authors also applied post-processing schemes to identify the rankings of significant terms, search for the global maximum and minimum values over the entire analytical zones, and perform optimization by considering the design cost. Table 2 summarizes the ANN models and preprocessing schemes applied to perform regression analysis. GO content (wt%), POSS content (wt%), salinity (ppm), and operating pressure (atm) were selected as the inputs to the ANN models, in accordance with the study conducted by Shams et al. [19,21]. This is because the inputs contributed significantly to the membrane's performance.

**Table 2.** ANN models and data scaling schemes.

| | | |
|---|---|---|
| Input components (4) | | $x_1$ = GO (wt%), (max, min) = (0.0125, 0)<br>$x_2$ = POSS (wt%), (max, min) = (1.2, 0)<br>$x_3$ = Salinity (ppm), (max, min) = (9750, 1296)<br>$x_4$ = Pressure (atm), (max, min) = (21.68, 6.78) |
| Output components (3) | | $y_1$ = Contact angle ($°$ degree), (max, min) = (71.5, 43.3)<br>$y_2$ = Salt rejection (%), (max, min) = (97.4, 15.3)<br>$y_3$ = Permeation flux, (L/m$^2$h), (max, min) = (17, 3) |
| ANN models | | Single hidden layer neural network<br>Three hidden layers neural network |
| Activation functions | | TanH function at the hidden layer<br>Linear/Sigmoid functions at the output layer |
| Data scaling | Input x | Normalized between −1 and 1<br>Standardized to z-scores |
| | Output y | Normalized between 0 and 1 |

The formulas of the activation functions used in this study, namely tanh function Equation (1), linear function Equation (2), and cosh function Equation (3), are listed below.

$$\tanh(x) = \frac{\sinh(x)}{\cosh(x)} = \frac{e^{2x} - 1}{e^{2x} + 1} \tag{1}$$

$$f(x) = mx + c \tag{2}$$

where $m$ is the slope of the linear line and $c$ is the y-intercept.

$$S(x) = \frac{1}{1 + e^{-x}} \tag{3}$$

The order of correlation coefficient $R$-values can be sorted to represent the ranking of significant terms in the normalized dataset. The $R$-values for each output ($y_1$, $y_2$, and $y_3$) with respect to the inputs ($x_1$, $x_2$, $x_3$, and $x_4$) are calculated using Equation (4):

$$R = \frac{n(\sum xy) - (\sum x)(\sum y)}{\sqrt{\left[n\sum x^2 - (\sum x)^2\right]\left[n\sum y^2 - (\sum y)^2\right]}}$$

(4)

the standard deviation, s, was calculated using Equation (5), while the MSE was determined by using Equation (6) below:

$$s = \sqrt{\frac{\sum (X_i - \overline{X})^2}{n-1}}$$

(5)

$X_i$ is the data, $\overline{X}$ is the mean of the data, and $n$ is the number of data points.

$$MSE = \frac{1}{n}\sum_{i=1}^{n}\left(X_i - \widehat{X_i}\right)^2$$

(6)

$\hat{X}_i$ is the predicted value.

## 3. Results and Discussion

### 3.1. Regression Analysis and Performances

Three ANN models were proposed to examine the performance of regression analysis. The results are summarized in Table 3, where each model's coefficient of determination ($R^2$) and Mean Square Error (MSE) are compared for evaluating the trained results. The $R^2_{Ave}$ denotes the average value taken from $R^2_{CA}$ for contact angle, $R^2_{SR}$ for salt rejection, and $R^2_{PF}$ for permeation flux. By comparing the $R^2_{Ave}$ and MSE, all the results of the three models fall within acceptable ranges for fitting criteria, with $R^2_{Ave} \geq 0.90$ (90%) and MSE $\leq 0.00199$ (6% variance). Table 3 shows the $R^2_{Ave}$ increased and the MSE dropped progressively, indicating the prediction approached the targets more closely as we altered the neural network model from a single hidden layer model (shallow neural network) to a three hidden layer (deep neural network) model. The deep neural network enables tweaking the decision boundaries to accommodate complex patterns of responses. The validation results imply that the fitting condition is balanced where variances and biases are around the targets. In addition, the selection of the activation function at the output layer from linear to sigmoid and the data scaling scheme applied to the inputs from linear type (input: from $-1$ to 1) to z-score also contributed to stabilizing the errors in the backpropagation and scaling down to narrow the jagged pattern of the input data.

The $R^2_{CA}$ for contact angle, the $R^2_{SR}$ for salt rejection, and $R^2_{PF}$ for permeation flux predicted by Model 3 were 0.970, 0.951, and 0.937, respectively, as tabulated in Table 3. The range of $R^2$ is comparable with the work done by Madaeni et al. [36], where the $R^2$ of the AAN model with a 4–11 network architecture, log-sigmoid activation function was 0.94 in predicting the permeate flow. Similarly, Jawad et al. [37] reported an $R^2$ value of 0.973 in predicting the membrane flux by an ANN architecture of 9-25-24-40-1, log-sigmoid, tan-sigmoid, and a log-sigmoid activation function. Jawad et al. [38] developed the ANN model by using the osmotic pressure difference, feed solution (FS) velocity, draw solution (DS) velocity, FS temperature, and DS temperature as the inputs to predict the membrane flux. The ANN architecture was 5-10-1 with a tan-sigmoid activation function, and the $R^2$ was found to be 0.980. Nevertheless, the single-layer model (model 1 in Table 3) is sufficient to simulate the desalination process by considering factors such as simplicity to derive mathematical equations and adaptability to fit broader ranges for testing.

**Table 3.** ANN regression analysis performances.

| Model No | Hidden Layer | Activation Functions | Data Scaling | | R² of y Response | | | | MSE |
|---|---|---|---|---|---|---|---|---|---|
| | | | | | $R^2_{CA}$ | $R^2_{SR}$ | $R^2_F$ | $R^2_{Ave}$ | |
| 1 | Single: 8 neurons + 1 bias | Hidden: tanh Output: linear | Input: $[-1, 1]$ Output: $[0.1, 0.9]$ | T | 0.951 | 0.897 | 0.906 | 0.918 | 0.00186 |
| | | | | s | 0.02736 | 0.03640 | 0.04648 | - | - |
| | | | | V | 0.943 | 0.908 | 0.849 | 0.900 | 0.00187 |
| | | | | s | 0.02838 | 0.03635 | 0.05052 | - | - |
| 2 | Three: 11:8:4 neurons +1 bias each | Hidden: tanh Output: sigmoid | Input: $[-1, 1]$ Output: $[0.1, 0.9]$ | T | 0.959 | 0.957 | 0.961 | 0.959 | 0.00098 |
| | | | | V | 0.949 | 0.960 | 0.953 | 0.954 | 0.00095 |
| 3 | Three: 11:8:4 neurons +1 bias each | Hidden: tanh Output: sigmoid | Input: z-score Output: $[0.1, 0.9]$ | T | 0.976 | 0.950 | 0.953 | 0.959 | 0.00097 |
| | | | | s | 0.02049 | 0.02359 | 0.01879 | - | - |
| | | | | V | 0.970 | 0.951 | 0.937 | 0.952 | 0.00100 |
| | | | | s | 0.02218 | 0.02328 | 0.01916 | - | - |

Note: T: Training set, V: Validation set, and s: standard deviation. *CA*: Contact angle, *SR*: Salt rejection, *F*: Permeation flux, and *Ave*: Average.

Figures 1 and 2 show the correlation diagrams between the experimental and predicted data for the three outputs by a single-hidden layer (Model 1) and a three-hidden layer (Model 3), respectively. The charts show the variances and biases of the predicted points are well distributed over the linear dotted line or the targeted line. By comparing the distributions, the predicted values in Figure 2 moved closer to the targets with higher R² values compared to Figure 1, indicating the three-layer ANN can handle complex patterns of output responses [39]. Both diagrams for contact angle, Figures 1a and 2a, depict having the highest R² values compared with other outputs (Figure 1b,c and Figure 2b,c), implying input components predominate the output responses for salt rejection and permeation flux. This is explained by ranking the significant terms, as tabulated in Table 4.

Correlation *R* values are calculated in Table 4 to represent the significance level between the input and output components. The Table 5 matrix of each row was extracted for radar charts, as illustrated in Figure 3, to visualize the degree of significance of input parameters against each output response in descending order by labeling the arrows up ↑ and down ↓ to denote positive (proportional) and negative (inversely proportional) correlation signs, respectively.

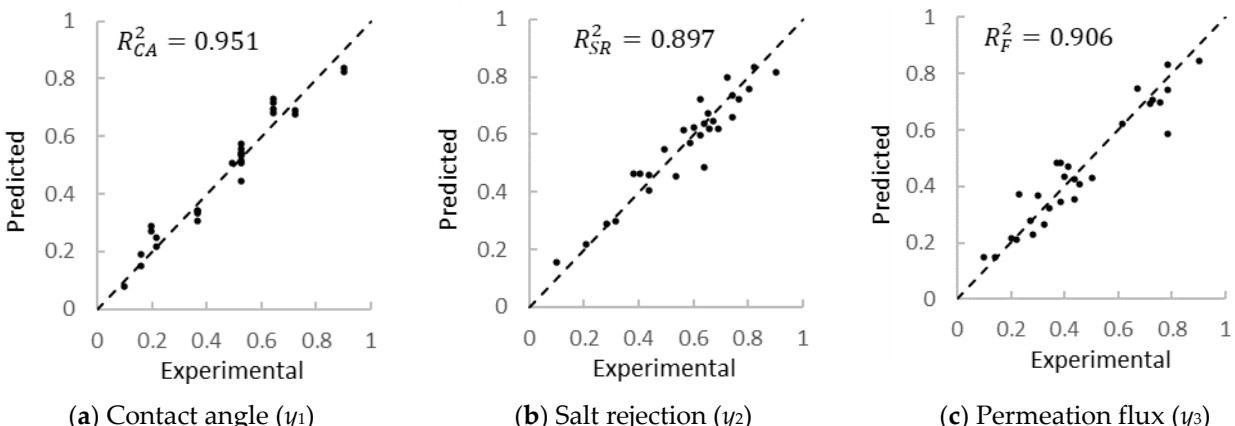

(**a**) Contact angle ($y_1$)     (**b**) Salt rejection ($y_2$)     (**c**) Permeation flux ($y_3$)

**Figure 1.** The goodness of fit tests for a single-layer ANN Model 1.

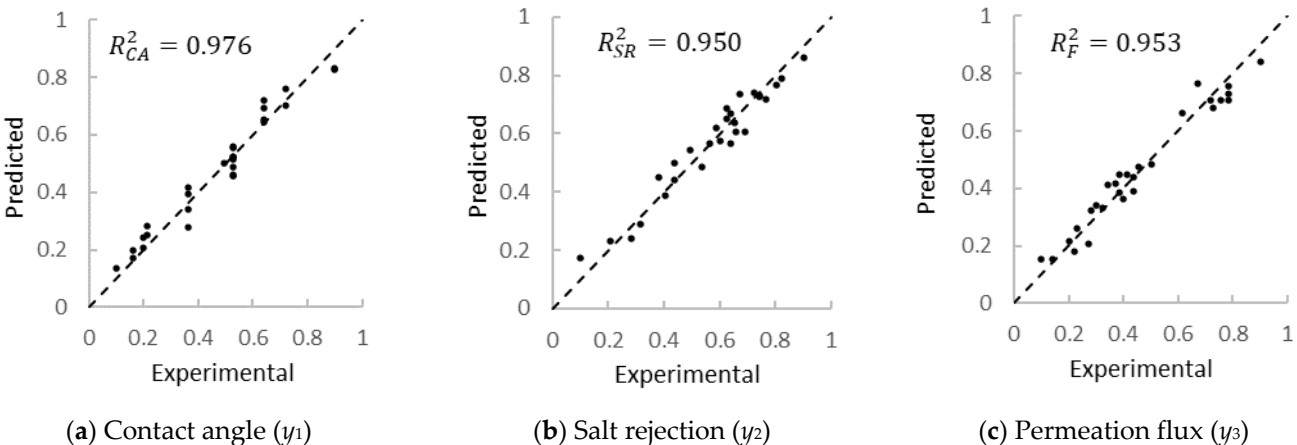

**(a)** Contact angle ($y_1$)    **(b)** Salt rejection ($y_2$)    **(c)** Permeation flux ($y_3$)

**Figure 2.** The goodness of fit tests for a three-layer ANN Model 3.

**Table 4.** Correlation $R$ between the input and output components.

| | | Correction R, Inputs | | | |
| | | **GO ($x_1$)** | **POSS ($x_2$)** | **Salinity ($x_3$)** | **Pressure ($x_4$)** |
|---|---|---|---|---|---|
| Output | Contact angle ($y_1$) | −0.8647 | −0.4228 | 0.1621 | 0.0010 |
| | Salt rejection ($y_2$) | −0.3053 | 0.4149 | −0.6360 | 0.0004 |
| | Permeation flux ($y_3$) | 0.2176 | 0.2698 | −0.2125 | 0.5410 |

**Table 5.** Global maximum and minimum outputs predicted from Equation (7).

| | | | Input | | | |
| | | | **GO** $x_1$ **(wt%)** | **POSS** $x_2$ **(wt%)** | **Salinity (ppm)** $x_3$ **(wt%)** | **Pressure (atm)** $x_4$ **(wt%)** |
|---|---|---|---|---|---|---|
| Output | CA $y_1$ (°) | Max = 69.14 | 0.0125 | 0 | 9705 | 21.68 |
| | | Min = 46.38 | 0 | 0 | 1296 | 14.18 |
| | SR $y_2$ (%) | Max = 93.11 | 0.0016 | 0 | 1296 | 6.68 |
| | | Min = 5.51 | 0.0125 | 0 | 9705 | 10.43 |
| | PF $y_3$ (L/m²h) | Max = 16.72 | 0.0125 | 0 | 1296 | 21.68 |
| | | Min = 1.12 | 0.0031 | 0 | 1296 | 6.68 |

Note: CA: Contact angle, SR: Salt rejection, PF: Permeation flux.

Figure 3a shows the ranking for the contact angle $y_1 : x_1 \downarrow > x_2 \downarrow > x_3 \uparrow > x_4 \uparrow$, indicating the GO content yields the strongest influence with $R = -0.864$, followed by POSS ($x_2$) with $R = -0.4228$. This finding is in accordance with the finding reported by the membranologists, where the hydrophilic additives contributed a significant impact to the membrane hydrophilicity, which was reflected by the contact angle value. Work done by Ooi and Chan [14] showed that the presence of iron oxide nanoparticles reduced the contact angle of the poly(vinylidene fluoride) (PVDF) membrane from 77° to 71°. Similarly, the presence of amphiphilic graphene oxide in poly(acrylonitrile) (PAN) composite membranes decreased the contact angle from 92.8° to 68.3° [40]. In this study, both POSS and GO served as inputs to the ANN model, while contact angle was one of the outputs. Thus, the strong relationship between the inputs and outputs was reflected by the high R value.

Meanwhile, Figure 3b shows the ranking for the salt rejection $y_2 : x_3 \downarrow > x_2 \uparrow > x_1 \downarrow > x_4 \uparrow$, indicating the salinity plays a substantive role despite the moderate R = −0.6360. Lastly, Figure 3c depicts the ranking for the permeation flux $y_3 : x_4 \uparrow > x_2 \uparrow > x_1 \uparrow > x_3 \downarrow$, indicating the pressure is an influential factor despite a moderate R = 0.5410. These findings suggest that salinity and operating pressure govern the salt rejection rate and permeation

flux, respectively. However, other factors, such as the pore size of the membrane and its thickness, which affect the membrane's performance, were excluded from the model. Hence, the $R^2$ values of salt rejection and permeation flux were slightly lower compared to the contact angle. This finding is in agreement with the work done by Wang et al. [41], where both salinity and pressure contributed significantly to salt rejection and permeation flux. It was because high pressure is required to overcome the osmotic pressure caused by a high salinity solution to ensure high salt rejection and permeation flux.

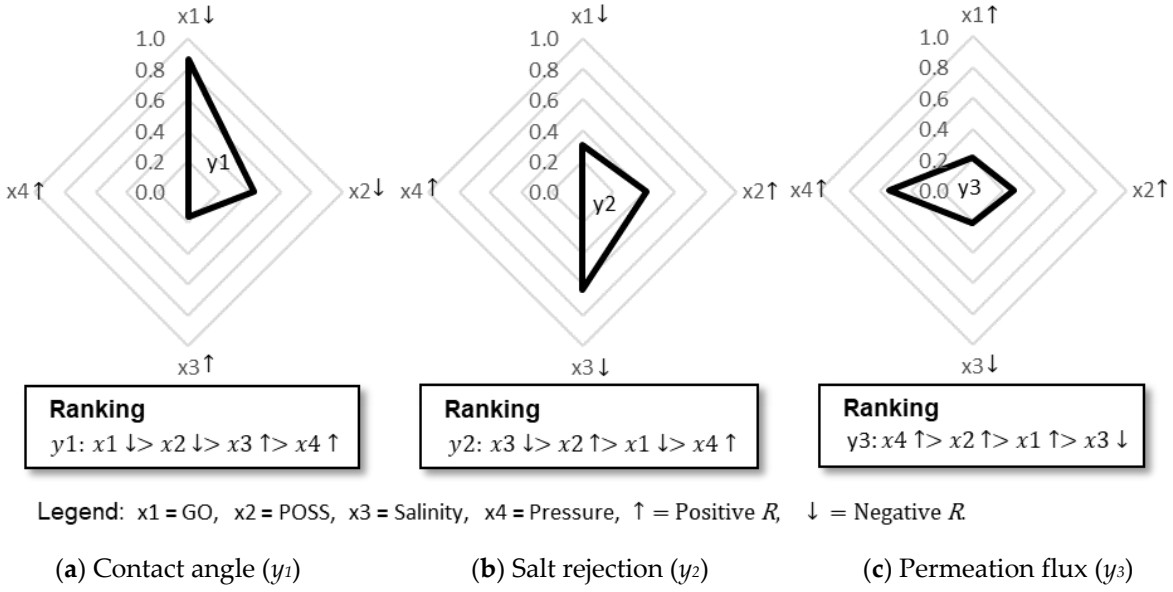

**Figure 3.** Ranking of significant terms by correlation.

### 3.2. Mathematical Equation of a Trained ANN Model

For practical reasons, engineers tend to look for convenient ways to get quick solutions by compromising precision with tolerance. Thus, a simple mathematical equation to instantly check the possible solutions at the planning stage of process design is desirable. After training a simple ANN model, we can derive an explicit mathematical equation $\{y_i\} = f(x_i)$ from the model for simulation and prediction applications, as shown in Equation (7). Equation (7) is derived based on the single-hidden layer model 1 in Table 3 to calculate contact angle ($y_1$), salt rejection ($y_2$), and permeation flux ($y_3$) from GO ($x_1$), POSS ($x_2$), salinity ($x_3$), and pressure ($x_4$).

$$
\begin{Bmatrix} y_1 \\ y_2 \\ y_3 \end{Bmatrix} = \begin{bmatrix} -0.0516 & -0.0474 & -0.1137 & -0.0966 & -0.4273 & 0.1014 & 0.0737 & -0.0034 & 0.3530 \\ 0.0345 & -0.4134 & -0.4981 & -0.1525 & 0.0751 & 0.3135 & -0.4154 & -0.0809 & 0.2508 \\ 0.0750 & 0.6107 & 0.0517 & 0.0321 & -0.0030 & -0.5284 & 0.6508 & 0.3387 & 0.6477 \end{bmatrix} \begin{Bmatrix} 2/\left(1+e^{-2x'_1}\right)-1 \\ 2/\left(1+e^{-2x'_2}\right)-1 \\ 2/\left(1+e^{-2x'_3}\right)-1 \\ 2/\left(1+e^{-2x'_4}\right)-1 \\ 2/\left(1+e^{-2x'_5}\right)-1 \\ 2/\left(1+e^{-2x'_6}\right)-1 \\ 2/\left(1+e^{-2x'_7}\right)-1 \\ 2/\left(1+e^{-2x'_8}\right)-1 \\ 1 \end{Bmatrix} \quad (7)
$$

where;

$$
\begin{Bmatrix} x'_1 \\ x'_2 \\ x'_3 \\ x'_4 \\ x'_5 \\ x'_6 \\ x'_7 \\ x'_8 \end{Bmatrix} = \begin{bmatrix} 0.0524 & 0.2340 & -0.0751 & -0.1497 & -0.0285 \\ -0.4280 & 0.9263 & -0.0026 & -0.0330 & -0.5070 \\ 0.2452 & -0.2172 & 0.4085 & -0.3937 & -0.4640 \\ 0.0456 & 0.0557 & -0.0812 & -0.1741 & 0.0715 \\ 0.5118 & 0.2789 & 0.0139 & 0.2449 & 0.2217 \\ -0.8437 & 0.4629 & 0.0449 & -0.1167 & 0.6888 \\ -0.2765 & -0.4459 & 0.3987 & 0.7951 & 0.6044 \\ -0.0851 & 0.1412 & -0.4668 & -0.2671 & 0.0170 \end{bmatrix} \begin{Bmatrix} x_1 \\ x_2 \\ x_3 \\ x_4 \\ 1 \end{Bmatrix} \tag{8}
$$

Figure 4 shows the predicted results calculated by Equation (7) compared with the experimental data. The postprocessing procedure is applied to revert the data scaling from a normalized unit $\{x, y\}$ to the original unit $\{\tilde{x}, \tilde{y}\}$ with the following matrix calculation, Equation (8).

$$
\begin{Bmatrix} \tilde{x}_1 \\ \tilde{x}_2 \\ \tilde{x}_3 \\ \tilde{x}_4 \\ \tilde{y}_1 \\ \tilde{y}_2 \\ \tilde{y}_3 \end{Bmatrix} = \begin{bmatrix} 0.0063 \\ 0.6 \\ 4204.5 \\ 7.5031 \\ 35.25 \\ 102.625 \\ 17.5 \end{bmatrix} \begin{Bmatrix} x_1 \\ x_2 \\ x_3 \\ x_4 \\ y_1 \\ y_2 \\ y_3 \end{Bmatrix} + \begin{Bmatrix} 0.0063 \\ 0.6 \\ 5500.5 \\ 14.1804 \\ 39.775 \\ 5.0375 \\ 1.25 \end{Bmatrix} \tag{9}
$$

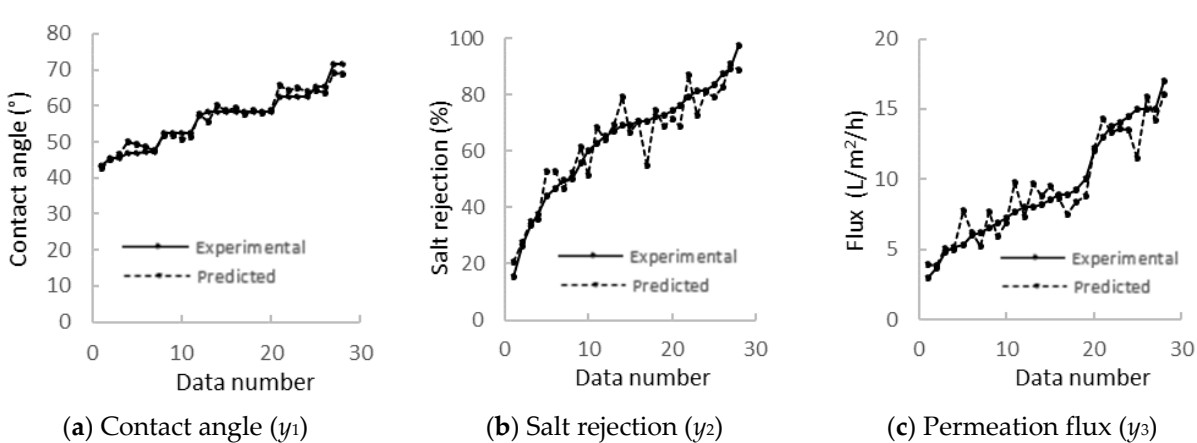

**(a)** Contact angle ($y_1$)      **(b)** Salt rejection ($y_2$)      **(c)** Permeation flux ($y_3$)

**Figure 4.** Regression analysis performance of the ANN from Equation (7).

The predicted contact angle data is closely aligned with the experimental data (Figure 4a), where some deviation was observed in the salt rejection (Figure 4b) and flux data (Figure 4c). This could be due to the higher sensitivity of the model to contact angles compared to salt rejection and flux. This is because the model was developed using POSS, GO, operating pressure, and salinity, where the R values of POSS and GO were higher for contact angle than the R values of salinity and operating pressure for salt rejection and flux. Figure 3 clearly demonstrates this. This finding is consistent with the data presented in Figure 1, where smaller deviations from the linear line were observed for contact angle data compared to salt rejection and flux data.

### 3.3. Three-Dimensional Response Patterns Simulated by the ANN Model

A trained ANN model was applied to create 3D response surfaces. The goal is to visualize the surface topography in order to better understand the response characteristics. In this study, we focused on using GO ($x_1$) as membrane additives by excluding POSS ($x_2$) because it contributed the same effect to all the output and was the second dominant factor

compared among all the four inputs. This is also to reduce the space dimensions from 4D to 3D in Table 4 to enable plotting 3D charts. From the rankings in Figure 3, we selected the top two influential input variables to form the input xy-plane with discrete points generated between −1 and +1 and plotted a response surface in the output z-plane by assigning a value to the remaining third input variable as a slicer, as illustrated in Figure 5.

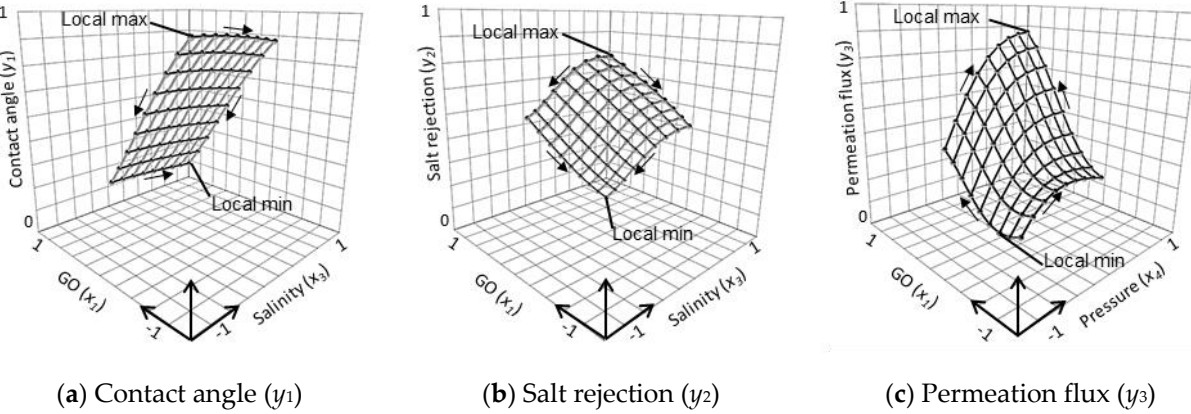

(**a**) Contact angle ($y_1$)  (**b**) Salt rejection ($y_2$)  (**c**) Permeation flux ($y_3$)

**Figure 5.** Single-layer response surfaces simulated by the ANN model.

Figure 5 shows the 3D polygon meshes representing response surfaces calculated by Equation (7). The meshes were drawn using the MS Charts 3D software, in which nodal points and lines represent the joints and boundaries of polygon elements. Based on the ranking order in Figure 3, GO ($x_1$) and salinity ($x_3$) are the influential inputs chosen as plane variables to create contact angle ($y_1$) meshes in Figure 5a. In this case, the pressure ($x_4$) variable is fixed at the middle point, $x_4 = 0$, to slice the 3D plane. Similarly, the same input variables and slicer were used to generate salt rejection ($y_2$) meshes in Figure 5b. On the other hand, GO ($x_1$) and pressure ($x_4$) are selected to plot permeation flux responses ($y_3$) in Figure 5c, where salinity ($x_3$) is fixed at the middle point $x_3 = 0$ as the plane slicer. Figure 5a shows the contact angle ($y_1$) forms a flat surface over the GO ($x_1$) and salinity ($x_3$) planes. A parabolic surface mesh with a negative inclined slope is created in Figure 5b for the salt rejection ($y_2$) over the GO ($x_1$) and salinity ($x_3$) plane, while a hyperbolic paraboloid mesh with a positive inclined slope is observed for the permeation flux ($y_3$) over the GO ($x_1$) and pressure ($x_4$) plane in Figure 5c. We can rotate the 3D chart to examine the topography and identify the locations of the local maximum and minimum points.

Figure 5b,c are similar to the 3D response surface plots published by Shams et al. [19]. Maximum permeation was obtained when 1 wt% GO was used to fabricate the membranes and the separation process was operated at 18 bar. It is notable that the relationship between pressure and flux was linearly proportional to each other as reported by Shams et al. [19], but the relationship was a nonlinear parabolic function when it was analyzed using ANN, as indicated in Figure 5c. It is because ANN can fit the data in a more flexible form compared to RSM [42].

Three-dimensional wireframe blocks, which envelope all the possible solutions within, were built through the layering method by toggling the slicer with specific increments moving from minimum to maximum. In this study, five incremental steps [−1, −0.5, 0, +0.5, and +1] were assigned to the slicer to generate five layers using Equation (7), as shown in Figure 6. Compared with Figure 5a for the contact angle ($y_1$), Figure 6a illustrates the layers being stacked to form a laminar flat plate. Meanwhile, for salt rejection in Figure 6b, the different gradient parabolic surfaces layered to create hump blocks intersect near the low salinity range ($y_2$). In Figure 6c, a complex saddle object layered by hyperbolic surfaces is seen for permeation flux ($y_3$), with intersections appearing from the middle to the low pressure. Since all the possible solutions are explicitly displayed with the blocks, we can examine the objects to identify the locations and values of the global maximum and global minimum, as labeled in Figure 6. The findings are summarized in Table 5. This 3D wireframe could

be used as a prediction chart for engineers, especially when there is a fluctuation in the salinity of seawater in practical applications. The engineers can easily estimate the salt rejection rate by using the 3D wireframe blocks.

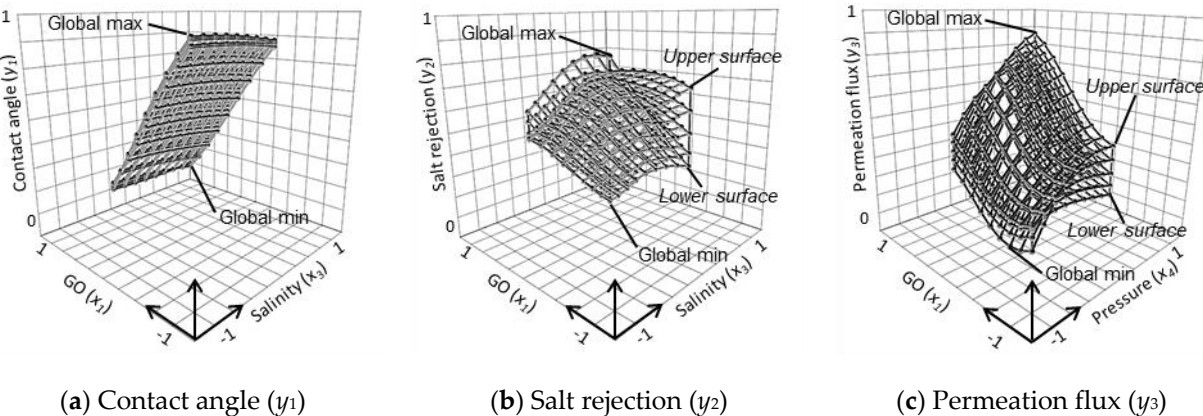

(**a**) Contact angle ($y_1$)    (**b**) Salt rejection ($y_2$)    (**c**) Permeation flux ($y_3$)

**Figure 6.** Three-dimensional wireframe block response patterns simulated by the ANN model.

*3.4. Optimizing Desalination Performance*

A design conflict exists in the desalination performance where an attempt to maximize the salt rejection rate leads to the dropping of permeation flux when comparing Figure 5b with Figure 5c. This is one of the major challenges faced by the membranologists: the membrane, which exhibits a high solute rejection rate, is always accompanied by low flux behaviors [43–45]. In order to maintain a high permeation flux, Figure 5c reveals that higher GO content and higher pressure are required to counter the drop, leading to cost burdens to run the operation. Thus, it is important to identify a balancing state to maximize performance.

The existence of optimized solutions can be proven by pairing the output responses with respect to each input, as shown in Figure 7. Figure 7a shows how the salt rejection ($y_2$) and permeation flux ($y_3$) outputs react uniquely against the GO content input ($x_1$) by stationing other inputs at $x_3 = x_4 = 0$. The two curves vary in opposite trends and intersect at around $x_1 = 0.1$, indicating a local optimum exists at this point if we attempt to maximize both outputs. A similar trend is observed in Figure 7b,c with respect to the salt rejection input $x_3$ and the pressure input $x_4$, where local optima can be determined by maximizing the objective functions (Equation (9)) subject to operating cost constraints.

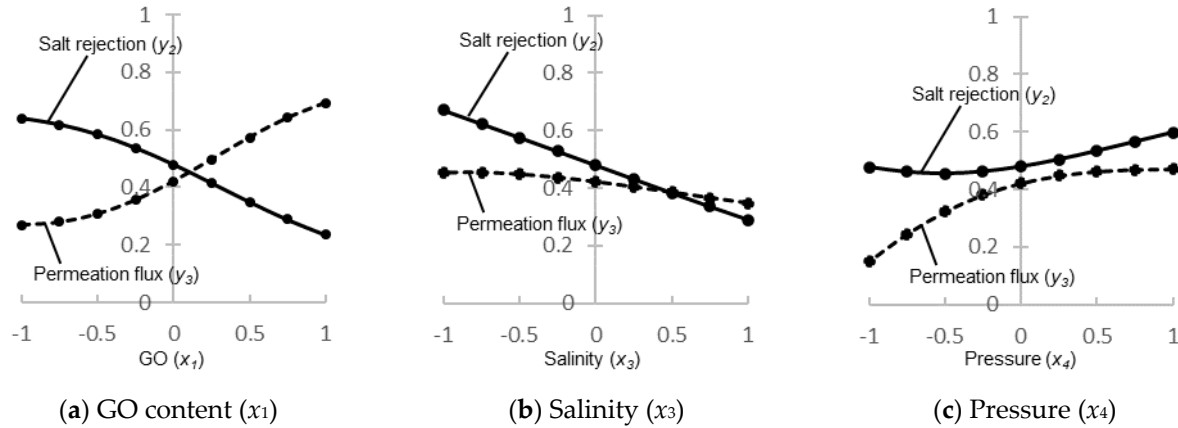

(**a**) GO content ($x_1$)    (**b**) Salinity ($x_3$)    (**c**) Pressure ($x_4$)

**Figure 7.** Output responses with respect to each input.

*3.5. Performance Criteria and Cost Constraints for Optimization*

With the output solutions calculated by Equation (7) as plotted in Figure 6, we can search for optimized performances by defining the objective function ∅ in terms of contact

angle ($y_1$), salt rejection ($y_2$), and permeation flux ($y_3$) subjected to input bounds, GO ($x_1$), POSS ($x_2$), salinity ($x_3$), and pressure ($x_4$) as follows:

$$
\begin{aligned}
\text{Maximize} \quad & \varnothing = \omega_1 y_1 + \omega_2 y_2 + \omega_3 y_3; \ \{y | 0 \le y \le 0.9\} \\
\text{where} \quad & \sum \omega_i = 1; \ i = 1, 2, 3; \ \{\omega | 0 \le \omega \le 1\} \\
\text{subject to} \quad & -1 < x_1 \le x_1^U, \ x_2 = -1, \ x_3^L \le x_3 < 1 \\
& -1 < x_4 \le x_4^U; \ \{x | -1 \le x \le 1\}
\end{aligned}
\tag{10}
$$

here $\omega_1$, $\omega_2$, and $\omega_3$ are the weights applied to penalize the respective output $y_i$ to target a specific objective. $x_1^U$ and $x_4^U$ are the upper bound variables, whereas $x_3^L$ is the lower bound variable associated with the operating cost constraints. $x_2 = -1$ indicates POSS content was excluded from the optimization. The analysis was made using the data in Table 1. Note that the terms expressed in Equation (9) use the normalized data set obtained from Equation (7). $\left(x_1^U, \ x_3^L, \text{ and } x_4^U\right)$ are the three input bounds set to confine a search region for finding the max $\varnothing$ value. $x_1^U$ and $x_4^U$ are the upper bound variables to limit the GO content and pressure (see Figure 6c) because lower is better to save the operating cost, whereas $x_3^L$ is the lower bound variable to hold the salt rejection rate (see Figure 6b) as high as possible to boost the desalination performance. In order to achieve the most cost-effective separation, it is desired to keep the concentration of GO in the membrane at a minimum ($x_1 = \sim -1$) and treat the water with the highest salinity ($x_3 = \sim 1$) at the lowest pressure ($x_4 = \sim -1$).

Equation (9) informs us that the objective function $\varnothing$ and the input bounds $x_i$ are the design elements that one can associate with performance criteria and cost constraints. By adjusting the weight $\omega_i$ in Equation (9), we can define three criteria ($\varnothing^1$, $\varnothing^2$, and $\varnothing^3$) to target specific performances, as described in Table 6. Here, $\omega_1 = 0$ is a penalty for disregarding the contact angle $y_1$. Contact angle served as one of the indicators of the antifouling property of a membrane [15]. It would affect the life span of a membrane, which eventually contributed to the operating cost [46]. However, the relationship of the contact angle to the membrane life span has not been studied systematically elsewhere. According to the review paper published by Coutinho de Paula and Amaral [47], an average RO membrane lifespan of six years was assumed to estimate the disposal rate of RO membranes annually. Thus, the following cost estimation was made by assuming the membrane cost is constant and that it is negligible when the daily operation cost is considered.

**Table 6.** Criteria to target different performances.

| Case | Criteria | Objective Function $\varnothing = \omega_1 y_1 + \omega_2 y_2 + \omega_3 y_3$ |
|---|---|---|
| 1 | Balanced outputs | $\varnothing^1 = 0 \cdot y_1 + 1/2 \cdot y_2 + 1/2 \cdot y_3$ |
| 2 | Higher salt rejection output | $\varnothing^2 = 0 \cdot y_1 + 2/3 \cdot y_2 + 1/3 \cdot y_3$ |
| 3 | Higher permeation flux output | $\varnothing^3 = 0 \cdot y_1 + 1/3 \cdot y_2 + 2/3 \cdot y_3$ |

By discretizing the interval $[-1, 1]$ into seven points ($-0.75$, $-0.5$, $-0.25$, 0, 0.25, 0.5, and 0.75) assigned to $\left(x_1^U, \ x_3^L, \text{ and } x_4^U\right)$, we can introduce a qualitative scale 'effective cost' to classify the operating cost from higher to lower by seven labels (H3, H2, H1, M, L1, L2, and L3), where point M is pinned to the zero middle point of the interval from $-1$ to 1. The rubric for mapping the effective cost with input bounds is shown in Table 7. The actual cost is not considered in the optimization procedure as the objective of this research is confined to analyzing the trends and determining the best solution.

**Table 7.** Rubric for effective cost and input boundary mapping.

| | Effective Cost | | | | | | |
| | Lower ← Middle ← Higher | | | | | | |
| **Input Bounds\Classes** | **L3** | **L2** | **L1** | **M** | **H1** | **H2** | **H3** |
| GO ($x_1^U$) | −0.75 | −0.5 | −0.25 | 0 | 0.25 | 0.5 | 0.75 |
| Salinity ($x_3^L$) | 0.75 | 0.5 | 0.25 | 0 | −0.25 | −0.5 | −0.75 |
| Pressure ($x_4^U$) | −0.75 | −0.5 | −0.25 | 0 | 0.25 | 0.5 | 0.75 |

Using the advanced filter tool in MS Excel, a criteria range table is prepared for a user to input combination values for $\omega_i$ from Table 6 and $\left(x_1^U, x_3^L, \text{ and } x_4^U\right)$ from Table 7. MS Excel will then compute the objective function $\varnothing$ and display a list of filtered results satisfying the criteria range. The sort tool is applied to sort the list in descending order to find the max $\varnothing$ solution.

The findings in Table 8 showed the optimized conditions ranged from high (H3) to low cost (L3) to meet the moderate salt rejection and permeation flux (case 1, $max\varnothing^1$), high salt rejection (case 2, $max\varnothing^2$), and high permeation flux (case 3, $max\varnothing^3$) requirements defined by the users. The normalized data are then reverted to their original units by Equation (8) to show the actual data from experiments in brackets. For instance, if it is desired to achieve the highest salt rejection (case 2, $max\varnothing^2$) at moderate cost (M), 0.016 wt% of GO is recommended to add into the membrane matrix to treat water with a salinity of 5501 ppm at 14.2 atm. A membrane with contact angle value of 66.7°, 68.4% of salt rejection rate, and 6.2 L/m²h are expected as the outputs.

To examine the significance of optimization by the three cases, the salt rejection $y_2$ and permeation flux $y_3$ output data from Table 8 are plotted by effective cost classes to compare the output performances as shown in Figure 8.

**Table 8.** Optimized solutions obtained from the interaction of the three cases defined in Table 6 and the seven classes described in Table 7.

| Case | Class | GO $x_1$ ($\tilde{x}_1$) (wt%) | POSS $x_2$ ($\tilde{x}_2$) (wt%) | SN $x_3$ ($\tilde{x}_3$) (ppm) | PS $x_4$ ($\tilde{x}_4$) (atm) | CA $y_1$ ($\tilde{y}_1$) (°) | SR $y_2$ ($\tilde{y}_2$) (%) | PF $y_3$ ($\tilde{y}_3$) (L/m²h) |
|---|---|---|---|---|---|---|---|---|
| | H3 | 0.75 (0.0109) | −1 (0) | −0.75 (2347) | 0.75 (19.8) | 0.29 (50.1) | 0.47 (53.5) | 0.75 (14.3) |
| | H2 | 0.5 (0.0094) | −1 (0) | −0.5 (3398) | 0.5 (17.9) | 0.37 (52.8) | 0.48 (54) | 0.68 (13.1) |
| | H1 | 0.25 (0.0078) | −1 (0) | −0.25 (4449) | 0.25 (16.1) | 0.45 (55.7) | 0.48 (54.5) | 0.58 (11.4) |
| 1 $max\varnothing^1$ | M | 0 (0.0063) | -1 (0) | 0 (5501) | 0 (14.2) | 0.54 (58.8) | 0.48 (54.3) | 0.42 (8.6) |
| | L1 | -0.75 (0.0016) | -1 (0) | 0.25 (6552) | −0.25 (12.3) | 0.76 (66.6) | 0.54 (60.8) | 0.18 (4.4) |
| | L2 | −0.75 (0.0016) | −1 (0) | 0.5 (7603) | −0.5 (10.4) | 0.75 (66.4) | 0.46 (52.5) | 0.21 (4.9) |
| | L3 | −0.75 (0.0016) | −1 (0) | 0.75 (8654) | −0.75 (8.6) | 0.74 (66) | 0.38 (44.3) | 0.21 (4.9) |
| | H3 | −0.75 (0.0016) | −1 (0) | −0.75 (2347) | 0.75 (19.8) | 0.75 (66.2) | 0.76 (83.5) | 0.38 (7.9) |
| | H2 | −0.75 (0.0016) | −1 (0) | −0.5 (3398) | 0.5 (17.9) | 0.76 (66.5) | 0.73 (79.8) | 0.34 (7.3) |
| | H1 | −0.75 (0.0016) | −1 (0) | −0.25 (4449) | 0.25 (16.1) | 0.76 (66.7) | 0.68 (74.8) | 0.31 (6.7) |
| 2 $max\varnothing^2$ | M | −0.75 (0.0016) | −1 (0) | 0 (5501) | 0 (14.2) | 0.76 (66.7) | 0.62 (68.4) | 0.28 (6.1) |
| | L1 | −0.75 (0.0016) | −1 (0) | 0.25 (6552) | −0.25 (12.3) | 0.76 (66.6) | 0.54 (60.8) | 0.25 (5.6) |
| | L2 | −0.75 (0.0016) | −1 (0) | 0.5 (7603) | −0.5 (10.4) | 0.75 (66.4) | 0.46 (52.5) | 0.21 (4.9) |
| | L3 | −0.75 (0.0016) | −1 (0) | 0.75 (8654) | −0.75 (8.6) | 0.74 (66) | 0.38 (44.3) | 0.17 (4.3) |

**Table 8.** *Cont.*

| Case | Class | GO $x_1$ $(\tilde{x}_1)$ (wt%) | POSS $x_2$ $(\tilde{x}_2)$ (wt%) | SN $x_3$ $(\tilde{x}_3)$ (ppm) | PS $x_4$ $(\tilde{x}_4)$ (atm) | CA $y_1$ $(\tilde{y}_1)$ (°) | SR $y_2$ $(\tilde{y}_2)$ (%) | PF $y_3$ $(\tilde{y}_3)$ (L/m²h) |
|---|---|---|---|---|---|---|---|---|
| | H3 | 0.75 (0.0109) | −1 (0) | −0.75 (2347) | 0.75 (19.8) | 0.29 (50.1) | 0.47 (53.5) | 0.79 (15.1) |
| | H2 | 0.5 (0.0094) | −1 (0) | −0.5 (3398) | 0.5 (17.9) | 0.37 (52.8) | 0.48 (54) | 0.68 (13.1) |
| | H1 | 0.25 (0.0078) | −1 (0) | −0.25 (4449) | 0.25 (16.1) | 0.45 (55.7) | 0.48 (54.5) | 0.55 (10.9) |
| 3 $max\varnothing^3$ | M | −0.25 (0.0047) | −1 (0) | 0 (5501) | 0 (14.2) | 0.62 (61.7) | 0.54 (60.1) | 0.36 (7.5) |
| | L1 | −0.25 (0.0047) | −1 (0) | 0.25 (6552) | −0.25 (12.3) | 0.62 (61.7) | 0.46 (52.6) | 0.31 (6.7) |
| | L2 | −0.5 (0.0031) | −1 (0) | 0.5 (7603) | −0.5 (10.4) | 0.69 (64.1) | 0.43 (49.2) | 0.23 (5.2) |
| | L3 | −0.75 (0.0016) | −1 (0) | 0.75 (8654) | −0.75 (8.6) | 0.74 (66) | 0.38 (44.3) | 0.17 (4.3) |

Note: SN: salinity, PS: pressure, CA: Contact angle, SR: Salt rejection, PF: Permeation flux.

One can observe in Figure 8 that cases 1 and 3 show similar trends, where the salt rejection output remains constant at around 0.48 or 54% from class H3 to H1 to boost higher permeation flux output and diverges slightly from class M to L3 to meet the respective criteria. Meanwhile, in case 2, the salt rejection output is higher than the permeation flux in all classes, indicating the optimized solutions have fully met the criterion.

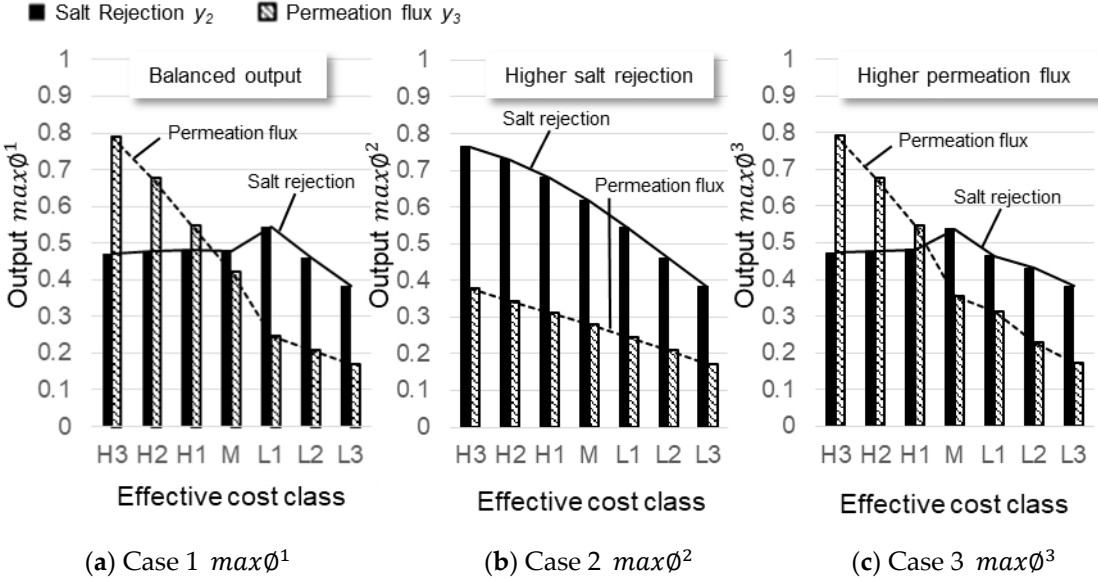

(**a**) Case 1 $max\varnothing^1$      (**b**) Case 2 $max\varnothing^2$      (**c**) Case 3 $max\varnothing^3$

**Figure 8.** Output performances for the three cases.

The data in Table 8 were mapped into 3D charts in Figure 6 to visualize how the optimized paths were captured. Figures 9 and 10 show the seven points from class H3 to L3 were joined to draw curves representing high-to-low optimized paths inside the solution blocks. The paths shown can be interpreted as the user's preference for optimal performance by factoring in the targeted outputs and cost constraints. By comparing the figures, one can see that the optimized paths varied subtly for each case, especially the path in case 2, which is unique from the others. For cases 1 and 3, both paths exhibit a similar trend from point H3 to point H1 and begin to split from point M in different directions. Meanwhile, the path for case 2 shows that the input GO content is bound at a lower value of −0.75 to maximize the salt rejection output. We can project the optimal paths onto the 3D charts in Figures 9 and 10 to understand how the objective functions work to find solutions.

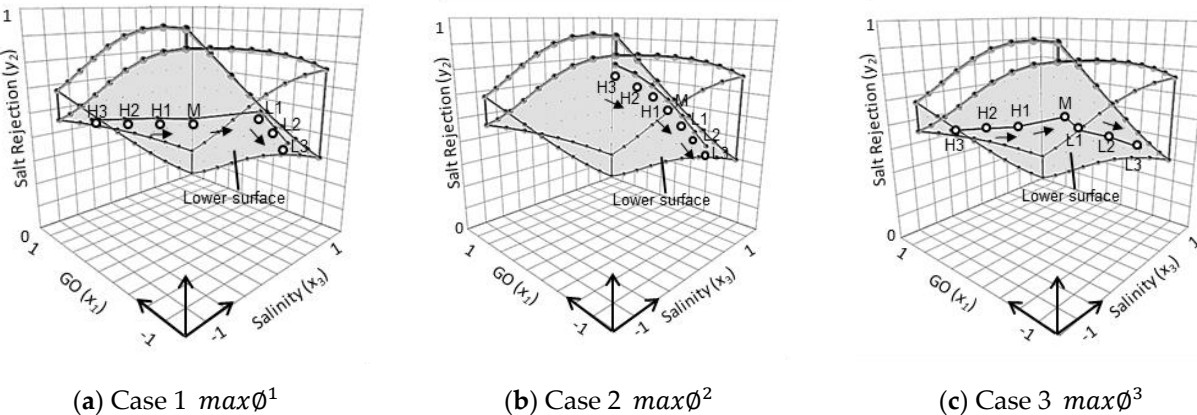

(**a**) Case 1 $max\emptyset^1$                      (**b**) Case 2 $max\emptyset^2$                      (**c**) Case 3 $max\emptyset^3$

**Figure 9.** Optimized paths for salt rejection output ($y_2$).

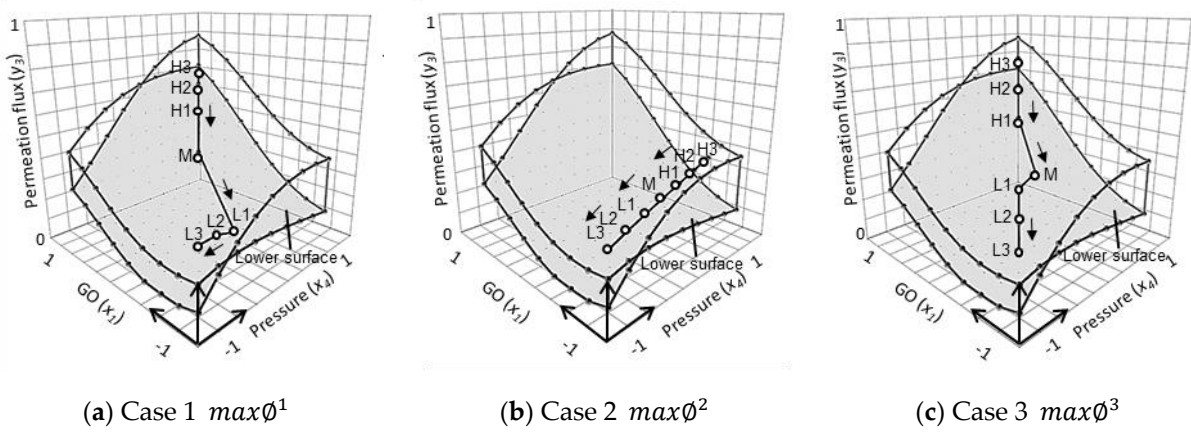

(**a**) Case 1 $max\emptyset^1$                      (**b**) Case 2 $max\emptyset^2$                      (**c**) Case 3 $max\emptyset^3$

**Figure 10.** Optimized paths for permeation output ($y_3$).

Considering a common practice in engineering design for striking a balance between the conflicting elements where one attempts to squeeze for higher output performance while suppressing the cost, the trends in Figures 9 and 10 reveal that the best solution is the one of case 1 point M from Table 8. This point is suggested based on the cost-effective perspective to emphasize equal outputs for salinity and permeation flux with a middle range of operating costs. A total of 0.0063 wt% of GO is required to be added to the polymer matrix to produce the desalination membrane with a contact angle of 58.8°. Under the operating pressure of 14.2 atm to treat 5501 ppm of NaCl, this membrane exhibits 54.3% salt rejection and 8.6 L/(m²h) of permeation flux.

## 4. Limitations and Recommendations

The accuracy of the ANN model was confirmed by high $R^2$ (>0.9) and low MSE (>0.002) values via the K-fold cross-validation scheme. It also compared and verified the RSM model developed in the previous study. In order to enhance the significance of the study, it is recommended to conduct experimental work to identify the difference between the predicted and actual membrane separation performances.

The ANN model is tailor-made for this GO/POSS/cellulose RO membrane under the specific range of salinity (1296–9750 ppm NaCl) and operating pressure (6.78–21.68 atm). The accuracy of ANN could be reduced if the input data are outside the desired range. It is recommended to adopt a pattern recognition approach to provide the prediction as a function of probability.

### 5. Conclusions

The ANN method is applied to reconstruct the input parameters and output responses by merging the experimental data previously obtained from RSM analysis. Graphene oxide (GO) content, Polyhedral Oligomeric Silsesquioxane (POSS) content, operating pressure, and salinity were combined as input parameters for a four-dimensional regression analysis to predict the three responses: contact angle, salt rejection, and permeation flux:

1. The ANN analysis from shallow to deep layer models showed acceptable ranges of fitting criteria with $R^2_{Ave} \leq 0.90$ (90%) and MSE $\leq 0.00199$ (6% variance);
2. Correlation $R$ values were used to rank the significance level of input parameters against output responses. The rankings are sorted to show that GO content with $R = -0.8647$ and POSS content with $R = -0.4228$ have strong influences on contact angle, salinity ($R = -0.6360$ on the salt rejection), and operating pressure ($R = -0.5410$ on the permeation flux);
3. Three objective functions and three-dimensional diagrams were applied to optimize effective cost conditions. It served as the database for the membranologists to decide the amount of GO to be used to fabricate membrane by considering the effects of operating conditions such as salinity and pressure to achieve the desired salt rejection, permeation flux, contact angle, and cost;
4. The finding suggested that a membrane with 0.0063 wt% of GO, operated at 14.2 atm for a 5501 ppm salt solution, is the preferred optimal condition to achieve high salt rejection and permeation flux simultaneously.

**Author Contributions:** Conceptualization, M.C. and C.W.; methodology, M.C. and C.W.; software, C.W.; validation, C.W.; formal analysis, M.C. and C.W.; investigation, M.C., C.W. and P.L.; resources, A.S., Y.J. and S.A.M.; data curation, C.W.; writing—original draft preparation, M.C., C.W., P.L. and A.S.; writing—review and editing, M.C. and P.L.; visualization, C.W.; supervision, M.C.; project administration, M.C.; funding acquisition, M.C. All authors have read and agreed to the published version of the manuscript.

**Funding:** The research resources provided by SEGi University are greatly appreciated (SEGiIRF/2022-Q2/FoEBEIT/011).

**Data Availability Statement:** Not applicable.

**Conflicts of Interest:** The authors declare no conflict of interest.

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
