# Peer review of "Artificial Neural Network Model for Membrane Desalination: A Predictive and Optimization Study"

_computation, doi:10.3390/computation11030068_

Round 1

Reviewer 1 Report

The authors have carried out investigation on the study on: Prediction and Optimization of the Desalination Performance of Graphene Oxide/POSS/Cellulose Membrane using Artificial Neural Network, before acceptance the following Minor comments should be addressed.

Title: 1. Please rewrite the title in accordance with the theme of the paper.

Abstract: 1. Please avoid using abbreviations in the Abstract or use abbreviations with full form, mention it first time, then abbreviations can be used.

2. Please report the findings in the abstract, what is the enhancement?

Introduction: please check the grammar, typos throughout the introduction section, check the similarity index in the introduction section.

1. Please clearly add the Novelty statement at the end of the introduction section. Please add why the study is important and what are the outcomes of the study.

Material and Methodology:

1. Please add the standard deviation and uncertainty analysis of the study.

2. Take care of subscripts and superscripts in Tables and abbreviations.

Results and discussion

1. Please improve the overall R and D section, please add previous studies to support your claims.

2. Improve the quality of the graphs and add detailed comparisons between different materials. Does the stability of GO plays role? Please mention and add prediction model.

Conclusion:

Please check the future scope and add relevance of the study, it will be better if you can add the conclusion in points for better understanding.

Author Response

Kindly refer to the attachment. 

Reviewer 2 Report

1. The introduction of this paper only introduces the shortcomings of RSM model in seawater desalination. It is suggested to list more articles on other approximate models for comparison with the ANN model selected.

2. In the model introduction of "Materials and Methods", you can write out the formula of the model used, and at the same time, you can make more additions to the "MIKA.NN" involved.

3. There are many times in the article that the superscript and subscript format problem such as "(L/m2/h)" appears.

4. In "Regression analysis and performances", "RAve2 ≤ 0.90 (90%)" should be "RAve2 ≥ 0.90 (90%)".

5. In the Mathematical equation of a trained ANN model, "where smaller deviation from the linear line was observed for contact angle data, compared to salt rejection and flux data ". Have you analyzed the reasons for the large deviation of salt rejection and flux data?

6. The conclusion section of the paper should be listed in points.

Author Response

Kindly refer to the attachment

Reviewer 3 Report

The submitted article developed an ANN model to predict the membrane desalination performance by considering the factors which govern the membrane separation performance, namely GO content, POSS content, operating pressure, and Salinity. The overall organization and structure of the manuscript are appropriate. The paper is well-written, and the topic is appropriate for the journal.

Page 2, line 50, correct the formulas of aluminum oxide and silica.

Pages 2-3, lines 98-107, In the final paragraphs of the introductory section, the authors explain what is the core of their research. However, it has to be clearly stated by the authors what their contribution is that makes the research different enough in comparison to the other authors' works, and they have to further elaborate on the extent of novelty in their research. The novelty of the work must be more clearly demonstrated.

Page 3, write full names for Box-Behnken designs and Central composite design the first time where the abbreviations are mentioned,

Relate your results with existing literature to support your findings.

Page 5, include the goodness of fit tests for the developed models' performance testing,

Page 18, The conclusion section needs to be rewritten. Add on the main findings/results of the study. What is the main outcome based on the results? The authors should highlight this matter.

Pages 19-20, Check the reference list; the names of the journals should be given in abbreviation form.

Author Response

Kindly refer to the attachment
